# Bismuth-Based Oxyfluorides as Emergent Photocatalysts: A Review

**DOI:** 10.3390/molecules30183784

**Published:** 2025-09-17

**Authors:** Thomas Erbland, Sara Ibrahim, Lucas Pelat, Kevin Lemoine, Angélique Bousquet, Pierre Bonnet

**Affiliations:** Institut de Chimie de Clermont-Ferrand (ICCF), Université Clermont Auvergne, 24 Avenue Blaise Pascal, 63178 Aubière, France; thomas.erbland@uca.fr (T.E.); lucas.pelat@uca.fr (L.P.); kevin.lemoine@uca.fr (K.L.)

**Keywords:** bismuth oxyfluoride, photocatalysis, structure–property relationship, improvement of photocatalytic properties, heterojunctions, band gap engineering, oxygen vacancies, Aurivillius structure, CO_2_ reduction, visible-light activation

## Abstract

Bismuth-based oxyfluorides (BiO_x_F_3−2x_) have recently emerged as promising photocatalysts due to their unique electronic structures and tunable physicochemical properties. This review provides a comprehensive overview of these materials, focusing on their crystal structures, band gap characteristics, and photocatalytic performance. Particular attention is given to BiOF, Bi_7_O_5_F_11_, and β-BiO_x_F_3−2x_, highlighting the influence of fluorine’s high electronegativity and internal electric fields on charge separation and light absorption. The potential of Aurivillius-type oxyfluorides is also discussed. Structural modifications, such as the introduction of oxygen vacancies, morphology control, and metal/non-metal doping, are examined for their effects on photocatalytic efficiency. Furthermore, various synthesis techniques and heterojunction engineering strategies involving semiconductors, carbon-based materials, and metal nanoparticles are explored to improve light harvesting and reduce charge recombination. Applications in pollutant degradation and CO_2_ photoconversion are reviewed, demonstrating the versatility of these materials. Despite their promise, the challenges associated with phase identification and composition control are also emphasized, underlining the need for rigorous structural characterization. Future directions for optimizing the photocatalytic activity of bismuth-based oxyfluorides are outlined, focusing on strategies to enhance their performance.

## 1. Introduction

Energy shortages and environmental pollution are challenging issues that people are facing all over the world and which demand significant attention. Since its first report by Fujishima and Honda, photocatalysis has been considered a promising technology to solve such issues [1]. It could resolve the problems of solar energy harvesting, conversion, and storage. Solar energy can be converted into electrical and chemical energy through photoinduced redox reactions on photocatalysts. When a semiconductor photocatalyst is irradiated with photon energy equal to or greater than its band gap, the electrons (e^−^) are excited to the conduction band (CB), leaving holes (h^+^) in the valence band (VB). Some of these charges recombine, reducing the photocatalytic activity. However, other photogenerated e^−^/h^+^ pairs separate efficiently and migrate to the surface, inducing oxidation and reduction photocatalytic reactions, leading to pollutant degradation, water splitting, or CO_2_ photoconversion. This process can be attained when the potential of the created electrons and holes is appropriate for the reduction and oxidation species adsorbed on the surface of the photocatalyst. Moreover, the oxidation and reduction reactions depend tremendously on the photocatalyst used. Thus, the photocatalyst plays the most important role in determining the process efficiency. It not only determines the thermodynamics of the process through the crystal and electronic structure but it also determines the kinetics of photocatalysis. The surface reaction kinetics consist of light accumulation, charge separation, charge transport or recombination, and charge utilization. Hence, the photocatalytic performance can be enhanced by increasing the light harvesting, reducing the carrier recombination rate in the material bulk or surface, and increasing the specific surface area, thanks to doping, heterojunction construction, using a plasmonic effect, or playing with the microstructure and O-vacancies. Therefore, different classes of materials have been developed for photocatalytic applications. TiO_2_ has been widely used in a variety of environmental and energy photocatalytic reactions due to its high oxidizing and reductive potential, high stability, low cost, and low toxicity [2,3,4]. Nevertheless, TiO_2_ cannot satisfy the requirements for industrialization for many reasons. At this stage, photocatalysts have several disadvantages, such as low utilization of sunlight, low quantum efficiency, low recyclability, and low photocatalyst efficiency [5].

Recently, bismuth-based materials have attracted significant attention in different domains, such as optoelectronics [6,7], gas sensors [8,9], and photocatalysis [10,11]. For example, bismuth-based oxide semiconductors including Bi_2_O_3_ [12,13], BiVO_4_ [14], Bi_2_MoO_6_ [15], and Bi_2_WO_6_ [16] have relevant photocatalytic activities. Furthermore, pristine bismuth oxyhalides (BiOX, X = F, Cl, Br and I) have been widely studied as important V–VI–VII ternary semiconductors due to their unique layered structure and excellent physicochemical properties [17,18]. They crystallize into a tetragonal matlockite (PbFCl-type) structure and are built of interleaved fluorite-like [Bi_2_O_2_] slabs with double halogen slabs. Although much attention has been focused on bismuth oxyhalide photocatalysts, it has principally been centered on BiOCl, BiOBr, and BiOI more than BiOF [19,20,21,22]. Because of its high E_g_ of 4.2 eV, fluorinated bismuth derivative compounds were initially considered less interesting materials for photocatalytic applications compared to other bismuth-based halides, as extending light absorption into the visible range was the most widely explored strategy [19,20,21,22]. However, charge separation is also one of the crucial steps limiting photocatalytic activity. Recent studies have demonstrated that the formation of internal electric fields is a promising route to limit charge recombination [23,24]. Recently, bismuth oxyfluoride materials have attracted significant attention since fluorine has distinctive chemistry properties from those of chlorine, bromine, and iodine [19,20,21,22]. Due to its high electronegativity, fluorine can trap electrons firmly and creates internal electric fields, thus leading to an uneven electron distribution and promoting charge separation, thereby enhancing the photocatalytic efficiency [25,26,27]. Its wide band gap gives it higher oxidation and reduction potentials than those of other oxyhalides. Moreover, unlike other bismuth oxyhalides, BiOF exhibits a direct band gap, which leads to more efficient light absorption. Although the total number of studies on BiOF photocatalysts remains limited compared to those on other bismuth oxyhalides (1.9%) (Figure 1a), there has been a gradual increase in research activity since 2013, indicating emerging interest from the scientific community (Figure 1b).

To date, several excellent reviews have considered BiOCl, BiOBr, and BiOI materials from different perspectives, such as their activity sites and structural defects. Xiong et al. reported various methods for the controllable synthesis of bismuth-rich Bi_a_O_b_X_c_, in addition to different strategies for tailoring the photocatalytic behavior [28]. Ye et al. introduced the hybridization, facet effect, and photocatalysis mechanism for a visible-light-driven (VLD) BiOX photocatalyst with remarkably enhanced photocatalytic activity [29]. Jin et al. introduced the methods of morphology control, internal electric field adjustment, and surface modification for the synthesis of a highly efficient BiO_x_X_3−2x_ photocatalyst [30]. Regarding bismuth oxyfluorides (BiO_x_F_3−2x_), the only review available to date is that by Vinoth et al. (2024) [31], which focused exclusively on BiOF. However, bismuth oxyfluorides represent a broad family of compounds with varying O/F compositions (BiO_x_F_3−2x_) and crystalline phases, which display markedly different physicochemical properties. In the literature, a lack of rigor often results from the frequent confusion between BiOF and other BiO_x_F_3−2x_ phases, despite their fundamental differences. As a result, the current body of work remains sometimes inconsistent. A comprehensive and systematic review is therefore required to establish a clearer framework and provide solid foundations for future investigations into this family of photocatalysts. In this paper, we focus on recent advances for the development of significant bismuth-based oxyfluoride materials and their different photocatalytic applications. Firstly, we briefly discuss the unique crystal and electronic structures of known BiO_x_F_3−2x_ phases and, by extension, of bismuth-based oxyfluorides presenting Aurivillius structures. Then, we present potential applications and various strategies to enhance the photocatalytic activity of these compounds. Finally, we conclude with the existing challenges and prospects for achieving greener and more efficient photocatalytic applications using these photocatalysts.

## 2. Crystal and Electronic Structures of Bismuth-Based Oxyfluorides

Physicochemical characterization of bismuth oxyfluorides is of particular importance due to the structural complexity and diversity of the compositions reported in the literature. Among these compounds, BiOF is the most frequently described phase. However, for other compositions, it appears that their attribution is sometimes erroneous, due to confusion between all oxyfluoride and fluoride compounds, exhibiting distinct O/F ratios and crystal structures. This section aims to present in a comparative manner the main existing Bismuth oxyfluoride phases and the analytical methods used in the literature to characterize these materials. The objective is to clarify the criteria for reliable structural identification and to highlight recurring pitfalls encountered in the interpretation of experimental data.

The first studies attempting to describe the different phases of bismuth oxyfluorides were carried out by the Swedish chemist Bengt Aurivillius in the mid-20th century and focused on a description of the Bi_2_O_3_–BiF_3_ system [32,33,34]. Subsequently, the work of Antoinette Morell, through solid-state synthesis of oxyfluoride from BiF_3_ and Bi_2_O_3_ at 670 °C for 15 h under argon, made it possible to demonstrate the existence of a series of phases with the composition BiO_x_F_3−2x_ (0 ≤ x ≤ 1.5) [35]. Figure 2 summarizes this phase continuum, with particular emphasis on the phase domains described in Antoinette Morell’s solid-state study [35]. The colored horizontal stripes represent the phasic domains of the phases observed by Antoinette Morell et al. [35] (with α: 0 ≤ O/O + F ≤ 0.5; β: 0.16 ≤ O/O + F ≤ 0.26; β’: 0.26 ≤ O/O + F ≤ 0.33; γ: 0.59 ≤ O/O + F ≤ 0.67).

### 2.1. BiOF

BiOF is the most described bismuth oxyfluoride because of its structural similarity to other well-known BiOX photocatalysts (X = Cl, Br, I) [36]. Figure 3A shows the crystal structure of BiOX [37]. As determined from density functional theory (DFT), relaxed BiOCl, BiOBr, and BiOI show indirect band gaps. BiOF exhibits a direct or slightly indirect band gap, depending on whether Bi 5d states are considered or not [38]. The maximum band gap values of relaxed BiOF, BiOCl, BiOBr, and BiOI are 3.34, 2.92, 2.65, and 1.75 eV, respectively [38]. The width of the valence band increases gradually from relaxed BiOCl to BiOI. However, the width of the conduction band decreases from relaxed BiOF to BiOI. When Bi 5d states are considered, the estimated dipole moments of BiO_4×4_ polyhedra within the relaxed species are in the order of BiOF > BiOI > BiOBr > BiOCl [38]. Figure 3B shows the electronic band structures of the relaxed bismuth oxyhalide compounds [39].

BiOF exhibits a unique layered structure characterized by fluorite-like [Bi_2_O_2_]^2+^ slabs interleaved by double slabs of F^−^ ions. Similar to other bismuth oxyhalides, there is an internal electric field (IEF) between the [Bi_2_O_2_]^2+^ slabs and the anionic halogen layers, which can effectively induce efficient separation of the photoexcited charge carriers [17,18]. The stronger internal electric field, introduced by the higher electronegativity of the fluorine atoms in BiOF, can more efficiently reduce the recombination of photoelectron–hole pairs and enhance the photocatalytic performance of BiOX. Figure 4 shows the X-ray diffraction (XRD) pattern of BiOF. Its diffraction peaks appear at 14.2°, 27.8°, 28.7°, 33.8°, and 37.5° and correspond to the (001), (101), (002), (110), and (102) crystal planes of its tetragonal phase.

BiOF exhibits a phase transition under high pressures from a tetragonal PbFCl-type structure with the space group *P*4/*nmm* to an orthorhombic structure with the space group *Cmcm* at about 19.6 GPa [41]. Under pressure, these structures exhibit different predicted band gaps, 2.74 eV for *P*4/*nmm* and 2.47 eV for *Cmcm* [41]. At the transition pressure, the volume of BiOF experiences an obvious discontinuous change, characterizing its first-order nature [41].

### 2.2. Bi_7_O_5_F_11_

While Aurivillius had already synthesized the Bi_7_O_5_F_11_ monoclinic phase in 1955 and named it δ-BiO_x_F_3−2x_ [32], this structure was first determined by Laval et al. in 1994 after synthesizing it through the solid-state reaction of Bi_2_O_3_ and BiF_3_ at 290 °C [42]. Bi_7_O_5_F_11_ crystallizes in the monoclinic space group *C*2. Bi_7_O_5_F_11_ has an intricate three-dimensional framework of interconnecting Bi-centered polyhedra: Figure 5a,b presents the crystal structure and schematics of the fluorine–oxygen polyhedra around the Bi^3+^ cations in Bi_7_O_5_F_11_ [39]. In Figure 4, peaks in the Bi_7_O_5_F_11_ pattern appear at 17.2°, 26.7°, 28.2°, 29.3°, and 31.0° and correspond to the (111), (112), (311), (401), and (20-3) crystal planes.

Theoretical calculations based on the DFT method were performed by W. Hu et al. [43]. Figure 5c shows the band structure of Bi_7_F_11_O_5_. The tops of the valence bands and the bottoms of the conduction bands are located at the Z point, making Bi_7_F_11_ O_5_ a direct band gap semiconductor with a band gap value of 3.81 eV.

### 2.3. β-BiO_x_F_3−2x_

A face-centered cubic phase with a variable composition, named β-BiO_x_F_3−2x_ by Aurivillius, was first synthesized through the solid-state reaction of Bi_2_O_3_ and BiF_3_ at 670 °C [34]. β-BiO_x_F_3−2x_ crystallizes in the *Fm-*3*m* space group. Figure 3 presents the classically observed β-BiO_x_F_3−2x_ pattern. The peaks at 26.5°, 30.7°, 44.0°, 52.2°, and 54.7° correspond to the (111), (200), (220), (311), and (222) crystal planes of the cubic β-BiO_x_F_3−2x_ phase. Its lattice parameters being close to those of α-BiF_3_ make them easy to confuse using XRD.

Moreover, the modification of the O/F ratio in the β-BiO_x_F_3−2x_ phase leads to a very slight shift in the diffraction peak position and a variation in the intensity of the (200) to (111) peaks. A. Morell et al. determined that the monophasic domain of β-BiO_x_F_3−x_ synthesized through the solid-state reaction of Bi_2_O_3_ and BiF_3_ at 670 °C [35] is 0.41 ≤ x ≤ 0.62. Two β-BiO_x_F_3−2x_ specimens were synthesized through stirring and precipitation in ethylene glycol (EG) for 96 h at room temperature [44]. The composition can be modulated by varying the quantities of Bi^3+^ and F^−^ precursors (here, Bi(NO_3_)_3_.5H_2_O and NH_4_F). The crystal molecular formulas, extracted from cell parameters calculated through Rietveld XRD refinement, are BiO_0.54_F_1.92_ and BiO_0.85_F_1.30_ [44]. Figure 6 shows Zhan et al.’s XRD and Rietveld studies on the influence of the graphene oxide surface on the synthesis and structure of β-BiO_x_F_y_ [44]. Even if the synthesized oxyfluoride patterns agree well with the standard card for BiO_0.51_F_1.98_, the diffraction peaks shift slightly to a higher 2θ angle, indicating that the elemental ratio of O to F in the crystal gas increased, as it is associated with a decrease in the lattice parameters. Despite these studies, most authors have not determined the precise composition of their synthesized compounds and have only identified the compounds through an XRD comparison to the standard BiO_0.51_F_1.98_, described by A. Morell et al. [35].

The experimental band gap of BiO0.51F1.98 synthesized through the hydrothermal method was determined, with a band gap value Eg = 3.35 eV, ECB = 1.73 eV, and EVB = 5.08 eV [45].

### 2.4. The Less-Described BiO_x_F_3−2x_

BiOF, Bi_7_O_5_F_11_, and β-BiO_x_F_3−2x_ have been detailed most in the literature and have shown the most convincing photocatalytic performance among all bismuth oxyfluorides. However, other bismuth oxyfluoride phases with less-known characteristics are also described in the literature and are presented in this section.

#### 2.4.1. α-BiO_x_F_3−2x_

α-BiO_x_F_3−2x_ was first synthesized by Aurivillius and described as a tysonite-like structure indexing in the *P*6_3_*/mmc* space group with the composition BiO_0.1_F_2.8_ [32,34]. A. Morell et al. confirmed the hexagonal structure of α-BiO_x_F_3−2x_ and determined the lattice parameters of the composition BiO_0.14_F_2.72_ [35]. This structure does not exist with the composition BiF_3_ but is stabilized by the occurrence of vacancies in the non-metal framework.

#### 2.4.2. β′-BiO_x_F_3−2x_

β′-BiO_x_F_3−2x_ is an orthorhombic phase whose lattice parameters are derived from β-BiO_x_F_3−2x_. The standard composition is BiO_0.67_F1_1.66_, described by A. Morell et al. [35], with the monophasic domain being 0.62 < x ≤ 0.74. The experimental band gap of BiO_0.67_F_1.66_ was determined from BiO_0.67_F_1.66_ synthesized through the hydrothermal method with the band gap value E_g_ = 3.39 eV, E_CB_ = 1.48 eV, and E_VB_ = 4.87 eV [45].

#### 2.4.3. γ-BiO_x_F_3−2x_

A hexagonal BiO_x_F_3−2x_ phase, named γ by Aurivillius and described further by A. Morell, was synthesized through the solid-state method with the monophasic domain 1.11 ≤ x ≤ 1.20 (Figure 7a). The composition BiO_1.18_F_0.64_ (also referenced as Bi_50_O_59_F_32_) is used as the standard name in the literature.

#### 2.4.4. Other BiO_x_F_y_ Structures

An anion-excess fluorite defective structure derived from a rhombohedral LnFO type was synthesized through annealing at 500 °C for 12 h in gold sealed tubes and then water-quenched [46]. The structure was solved using XRD on a single crystal in the *R-*3*m* space group with the composition BiO_0.9_F_2.35_ (Figure 7b). It can be mentioned that this phase was later synthesized through the hydrothermal method by M. Ren et al., and its photocatalytic properties were tested for the degradation of RhB under UV light [47]. Considering the soft synthesis conditions compared to those of Laval et al. [46], it seems unlikely that it was BiO_0.9_F_2.35_ but perhaps a confusion with the cubic β phase, which has close diffraction peaks.

Centered cubic-phase Bi_26_O_38_F_2_ with a sillenite structure can be synthesized through precipitation from Bi(NO_3_)_3_·5H_2_O and NaF in NaOH solution [48]. This phase is not considered for its own photocatalytic properties but is known to be used in composite photocatalysts for the formation of heterojunctions, notably with BiO_x_I_y_, g-C_3_N_4_, or graphene oxide (GO) [49,50,51].

### 2.5. The Aurivillius Structure

The Aurivillius structure was described by B. Aurivillius in 1949 using the general formula (Bi_2_O_2_)(A_n−1_B_m_O_3n+1_), where A is a large metal cation in 12 coordination, such as Pb^2+^, Na^+^, K^+^, Ca^2+^, Sr^2+^, Ba^2+^, or Bi^3+^, and B is a transition metal in an octahedral coordination, such as Ti^4+^, Nb^5+^, Ta^5+^, Fe^3+^, Mo^6+^, or Ga^3+^ (with *n* ≤ 5) [33]. The structure is made up of fluorite layers [Bi_2_O_2_]^2+^, as shown in Figure 8a, similar to the slabs in the BiOF structure, and *n* alternating perovskite layers [A*_n_*_−1_B*_m_*O_3*n*+1_]^2−^ along the *c* axis, as for Bi_2_MoO_6_ and Bi_4_Ti_3_O_12_, with *n* = 1 and 3, respectively. Aurivillius structure oxides generally crystallize in the centrosymmetric tetragonal space group *I*4/*mmm*. Substitutions of oxygen by fluorine have allowed for the formation of oxyfluoride Aurivillius structures with distinctive electronic functionalities that induce interesting ferroelectric properties for applications in photocatalysis, as spontaneous polarization allows for an improved surface chemistry and charge separation, induced by an internal electric field.

A centrosymmetric tetragonal structure *I*4/*mmm* of the phases Bi_2_NbO_5_F, Bi_2_TaO_5_F, and Bi_2_TiO_4_F_2_ was proposed in 1952, considering total anionic disorder for the oxygen and fluorine atoms (Figure 8b), and a Bi_2_VO_5_F structure was resolved following the same model [55]. Later, the orthorhombic space group *Pbca* of a lower symmetry was proposed for Bi_2_NbO_5_F [56]. Based on calculations of the valence of the anions, the problem of the fluorine atom positions was difficult to elucidate at the level of their localization at the equatorial or apical sites of the Nb(O,F)_6_ octahedra. Preferential localization of fluorine atoms at the apical sites was ultimately demonstrated.

According to the literature, these three phases are ferroelectric, with transitions measured at Tc = 303, 283, and 284 K for Nb^5+^, Tb^5+^, and Ti^4+^, respectively [56]. The electronic structure of Bi_2_TiO_4_F_2_ has been investigated through DFT calculations by S. Wang et al. and shows that Bi_2_TiO_4_F_2_ exhibits an indirect band gap in the visible part with an energy of 1.46 eV. The band gap energy was also measured through diffuse reflectance spectroscopy and determined to be 3.26 eV [57]. J. Wang at al. investigated the electronic structure of Bi_2_NbO_5_F through DFT calculations and demonstrated that Bi_2_NbO_5_F is a direct semiconductor with a band gap energy of about 2.50 eV. The band gap energy was also measured experimentally through DRS, with a value of 3.14 eV [58]. Both studies show a larger experimental band gap energy due to underestimation of the band gap in the DFT calculations [59].

## 3. Challenges in Characterizing BiO_x_F_3−2x_

A major contributing factor to the confusion between the different bismuth oxyfluoride phases is the similarity of their XRD patterns, particularly the recurring presence of an intense peak around 27°. Its almost systematic presence complicates the precise identification of the phases and underlines the need for a more detailed analysis, combining XRD with other techniques for reliable attribution of the structures. Moreover, some crystalline phases in BiO_x_F_3−2x_ bismuth oxyfluorides have variable anionic compositions. Despite this, many studies only report compositions from bibliographic references, without really quantifying the O/F ratio, whereas this variation in the O/F ratio has a direct impact on the physicochemical properties of the material, particularly on its performance in photoelectrochemistry [45]. Rigorous determination of the actual composition is therefore essential to understand and optimize the functional properties of these compounds. Indeed, to avoid misinterpretation in structure–property correlations, complementary characterization techniques should be performed.

Figure 9a,b present the Raman and FTIR spectra for BiOF, Bi_7_O_5_F_11_, and β-BiO_x_F_3−2x_ materials. In BiOF’s Raman spectrum, the vibration bands at 169 and 241 cm^−1^ are attributed to the A_1g_ and E_g_ Bi-F stretching modes, respectively. In the Bi_7_O_5_F_11_ Raman spectrum, the weak peak at 595 cm^−1^ and the peak at 444 cm^−1^ can be assigned to ν(Bi-O) and ν(Bi-F) vibrations, respectively. The peaks below 400 cm^−1^ are attributable to the bending and stretching vibrations of Bi-O-Bi and Bi-F-Bi [43]. In the Raman spectrum of β-BiO_x_F_3−2x_, the band observed at 157 cm^−1^ can be attributed to Bi-O and Bi-F stretching vibrations, while the band at 313 cm^−1^ is attributed to the stretching and bending vibrations of Bi-O-Bi and Bi-F-Bi. Finally, the band at 449 cm^−1^ is attributed to Bi-O^−^. In the FTIR spectrum of BiOF, the bands observed at 420 and 550 cm^−1^ correspond to the Bi-O and Bi-F stretching vibrations in BiOF, respectively. In the FTIR spectrum of Bi_7_O_5_F_11_, the two bands at 604 and 526 cm^−1^ are assigned to the ν(Bi-O) and ν(Bi-F) vibrations of Bi_7_F_11_O_5_. Finally, in the FTIR spectrum of β-BiO_x_F_3−2x_ the weak bands at 485 and 545 cm^−1^ correspond, respectively, to the Bi-O and Bi-F stretching vibrations.

It appears that the characteristic bands of bismuth oxyfluorides in Raman and FTIR spectroscopy are associated with similar bonds, particularly Bi-F and Bi-O, but with shifted wavenumbers and different spectral profiles. Although these analytical techniques do not seem sufficient to clearly identify BiO_x_F_3−2x_, they may be relevant techniques to complement XRD characterization and provide further information on the compound.

Fortunately, the position of the Bi 4f peak in XPS is very sensitive to the presence of fluorine. Figure 9c presents the Bi 4f peak spectra for various bismuth oxyfluoride compounds. The Bi 4f peak is composed of a Bi 4f_7/2_ and Bi 4f_5/2_ doublet separated by 5.3 eV.

An XPS analysis might be particularly interesting for discriminating compounds presenting a diffraction pattern close to that referenced as β-BiO_0.5_F_2_ but that could have a composition ranging from Bi_2_O_3_ to BiF_3_. Indeed, fluorine’s high electronegativity highly influences the position of the Bi^3+^ 4f_7/2_ peak, which shifts from 158.7 eV for Bi_2_O_3_ to 160.0 eV for BiO_0.5_F_2_ and to 160.4 eV for BiF_3_ [60,61]. Moreover, the exact compound formula can be deduced from the quantitative elemental content analysis allowed by this technique.

Figure 9c also shows the Bi 4f spectra for BiOF, with the position of the Bi^3+^ 4f_7/2_ peak at 159.4 eV. Finally, we add the spectrum for Bi_7_O_5_F_11_. Since the latter has been studied very little, the exact position of its 4f_7/2_ peak is not referenced in the literature, but this position at 158.9 eV seems to be in agreement with the spectrum observed in [62].

Even if an XPS analysis does not give information on the crystalline structures of bismuth-based oxyfluorides, this technique could advantageously give insights into their O-to-F ratios. This could be useful, for instance, for obtaining the exact composition of a material for a structure close to BiO_0.5_F_2_.

## 4. Bismuth-Based Oxyfluorides as Photocatalysts and Strategies to Enhance Their Activity

### 4.1. Bismuth-Based Oxyfluorure Photocatalysts

Nowadays, the development of bismuth oxyfluoride as photocatalysts appears to be very promising. BiOF is the most widely studied bismuth oxyfluoride due to its photocatalytic properties. In the BiOX family, as shown in Figure 10a, as the halogen atomic number increases, the valence band position of BiOX gradually becomes more positive, giving BiOF a stronger oxidation potential. Therefore, BiOF is efficient for forming OH radicals from H_2_O oxidation [63,64]. It has therefore been tested for degrading organic pollutants such as methyl orange, methyl blue, Rhodamine B, Rhodamine 6G, ciprofloxacin, perfluorooctanoic acid, benzyl alcohol, nitrobenzene, crystal violet, 2-hydrobenzoic acid, and tetracycline. In the same way, the conduction band position of BiOX becomes more negative, also making BiOF suitable for activating reduction reactions for CO_2_ conversion into valuable molecules (CO, CH_4_…), with redox potentials around −0.106 (pH = 0) for CO_2_ to CO [65], for industrial CO_2_ emission abatement [37].

Besides BiOF, a wide range of BiO_x_F_3−2x_ materials with different atomic ratios have been synthesized and reported as promising photocatalysts. M. Ren et al. showed that many of these materials are easily accessible through hydrothermal synthesis by modifying synthesis parameters such as the reaction medium’s composition (Figure 10c) [45]. Compared to BiOF, lower-O/F-ratio BiO_x_F_3−2x_ materials present a broader and more positively positioned valence band, thereby enhancing their oxidative potential, as shown in Figure 10b. Furthermore, the different O/F ratios observed in compounds such as BiO_0.51_F_1.98_ (O/F = 0.25), Bi_7_O_5_F_11_ (O/F = 0.45), and BiO_0.67_F_1.66_ (O/F = 0.4) could be responsible for the variation in the intensities of their internal electric fields. As the O/F ratio decreases, oxygen defects tend to increase, which correlates with enhanced photocatalytic activity [45].

Concerning Aurivillius compounds, using a mixture of Bi(NO_3_)_3_·5 H_2_O and (NH_4_)_2_TiF_6_ [57] or H_2_TiF_6_ [66], nanoparticles and nanospheres of Bi_2_TiO_4_F_2_ have been synthesized through the solvothermal method. Both present good activities for photodegrading pollutants into water under UV–visible light. This strategy even allows porous spheres of Bi_2_M^5+^O_5_F (M = Nb and Ta) with photocatalytic activity to be formed [67]. Nanosheets of Bi_2_NbO_5_F were also synthesized through the molten salt method with preferential growth along the (010) plane, which could degrade tetracycline and RhB under UV twice as efficiently as the product synthesized through ceramic synthesis [58]. This synthesis was also adapted to using BiF_3_ instead of HF. The Bi_2_NbO_5_F nanosheets obtained present a photocatalytic activity 15.6 times higher than that of the compound obtained via the solid-state route because of a higher specific surface area [68].

The hydrothermal method can be extended to other Aurivillius compounds with a metal of oxidation number (+II), such as Bi_2_Co^2+^O_2_F_4_. Its crystal structure has been described in the tetragonal space group *I*4 with alternating [CoF_4_]_∞_ and [BiO_2_F_2_]_∞_ layers via sharing of fluorine atoms at the vertices [69]. The relatively weak interlayer Bi–F bonds offer the possibility of exfoliation through magnetic stirring in isopropanol into nanosheets, exhibiting a higher electrocatalytic activity for water oxidation than that of the initial material thanks to a larger specific surface area [70]. Finally, thin films with an Aurivillius-type structure were investigated. Using a (Bi_0.8_Ba_0.2_)FeO_2.9_ precursor and PVDF for top chemical fluorination at 350 °C, a (Bi_0.8_Ba_0.2_)FeO_2_F film is formed, with promising photocatalytic activity [71].

These findings highlight the value of going beyond the stoichiometric BiOF composition and finely tuning the stoichiometry of bismuth-based oxyfluorides to optimize their photocatalytic performance. Importantly, adjusting the crystal structure, the O/F ratio, or choosing to incorporate other metal cations into the Aurivillius structure provides a powerful strategy for tailoring the electronic structure and reactivity of these materials to meet the specific requirements of targeted photocatalytic applications.

To go further, different approaches to improving the photocatalytic performance of the pure compound have been considered, as illustrated by Figure 11. First, we may play with its morphology and surface to increase the surface reactivity and area. The material may also be doped with metal or non-metal elements to add new levels into the band gap to favor light absorption and carrier separation. Finally, it can be associated with carbon materials, another photocatalyst, or metal in the heterojunction, which favors light harvesting and reduces electron–hole recombination.

### 4.2. Facet Selection and Oxygen Vacancies

Oxygen vacancy defects have an important effect on the band structure and optical absorption properties of semiconductors. The existence of oxygen vacancies causes the Bi 6p state to fall into the forbidden band, which increases the effective separation of electron–hole pairs, redshifts the edge of the absorption band, and realizes an electronic transition from the F 2p or the O 2p state under visible-light irradiation [41]. Indeed, the presence of O-vacancies induces a decrease in the band gap, enhancing the light absorption, and contributes to carrier separation [73]. It is well known that a BiOF layered structure favors the presence of O-vacancies and accessibility between the [Bi_2_O_2_]^2+^ slabs. Zhang et al. [27] used a simple hydrothermal method to synthesize BiOF, where exposition of the {101} surface, rather than {001} surfaces, is controlled by the pH values. More surface O-vacancies are contained in the {101} nanodisk than common thick BiOF plates, which reduces the band gap and improves the charge transfer and separation, as illustrated in Figure 12. Therefore, the selection of a {101} facet, rather than a {001} facet, drastically improves the photocatalytic properties [27]. With this idea of facet engineering, Zou et al. used a simple hydrothermal method to produce BiOF nanosheets, where 75.4% of the high-energy (002) surface was exposed, showing a higher photocatalytic activity [26].

### 4.3. Metal/Non-Metal Doping

Doping, in the context of materials science, specifically refers to the deliberate incorporation of small quantities of foreign elements into the crystalline lattice of a material, achieved either through substitution or insertion. This distinction is often blurred in the literature concerning BiOF-based materials, where the concept of doping is sometimes confused with that of heterojunctions, defined as the interface formed between two crystalline materials of distinct compositions. In this section, we will present the effect of doping, understood as the incorporation of elements into BiO_x_F_3−2x_’s structure. The effect of heterojunction formation is discussed later.

Doping metal/non-metal ions into BiOF can accelerate the separation efficiency of photoelectron–hole pairs and hinder their recombination, in addition to adjusting the band gap. Table 1 below summarizes the work conducted on bismuth oxyfluoride doping and highlights the ubiquity of rare earth ions as an oxide doping element. Doping with these elements is indeed recognized for its effect on the band structure and a shift in the band gap towards the visible region [74]. Wang et al. demonstrated that Sm^3+^-activated BiOF nanoparticles show excellent photoluminescence properties as a visible-light-driven photocatalyst [75]. Vadivel et al. prepared Y-BiOF Reduced Graphene Oxide (RGO) composites and realized that the effective combination of yttrium ions and RGO sheets reduced the recombination rate of BiOF and improved the absorption of visible light [63]. Er^3+^/Yb^3+^-codoped BiOF nanoparticles with an enhanced photocatalytic activity were developed by P. Du et al. through a high-temperature solid-state reaction. The improved photocatalytic activity is linked to various effects: an enlarged absorption edge, the formation of impurity energy levels, and an enhanced upconversion performance, allowing for the degradation of RhB and MB under visible and NIR light, as shown in Figure 13 [76]. R. Saraf et al. synthesized Eu^3+^doped BiOF and evaluated its performance for RhB degradation under visible light [64]. The structural, electronic, and optical properties of Ag/Pd-doped BiOX (X = F, Cl, Br, I) were calculated by S. Zhou et al. based on DFT calculations. It was highlighted as standing out from other compounds due to its unusually broad absorption peaks, which could be beneficial in photocatalytic systems [77]. M. Elias et al. [78] synthesized BiOF, Ag-doped BiOF, and Ag-doped BiOF–rGO composites and tested their performance for the photodegradation of MB and MO under UV light. Ag-doped BiOF–rGO presents an enhanced photocatalytic performance due to a synergistic effect that promotes the efficient separation of photogenerated electron–hole pairs. It is proposed that the incorporation of Ag^+^ ions into the BiOF lattice improves the excitation rate, while rGO aids in the separation of e^−^/h^+^ pairs by directing them towards the interface, thus increasing the photocatalytic activity of the material [78].

**Table 1 molecules-30-03784-t001:** Synthesis methods, precursors, and photocatalytic applications of doped-BiO_x_F_3−2x_ materials.

Materials	Synthesis Method	Precursors	Application	References
Ag-doped BiOF–rGO	Solvothermal	GO, Bi(NO_3_)_3_·5H_2_O, AgNO_3_, NH_4_F, C_2_H_6_O_2_	The photodegradation of MB and MO dyes under UV-light and sunlight irradiation	[78]
Er^3+^/Yb^3+^-codoped BiOF	Solid-state	Bi_2_O_3_, NH_4_F, Yb_2_O_3_, Er_2_O_3_	The photodegradation of RhB and MB under visible-light and NIR-light irradiation	[76]
Y-BiOF/RGO	Solvothermal	Bi(NO_3_)_3_·5H_2_O, Y(NO_3_)_3_·6H_2_O, PVP mw40.000, C_2_H_6_O_2_	The degradation of MB under visible-light irradiation	[63]
Sm^3+^-doped BiOF	Solid-state	Bi_2_O_3_, NH_4_F, Sm_2_O_3_	The decomposition of RhB dye under visible-light irradiation	[75]
Eu^3+^doped BiOF	Solid-state	Bi_2_O_3_, NH_4_F, Eu_2_O_3_	The degradation of RhB dye under visible-light irradiation	[64]

### 4.4. The Formation of Heterostructures with Other Photocatalysts

Heterojunction engineering is a central strategy for enhancing the charge separation efficiency in photocatalytic systems. Based on the band structure alignment and interfacial charge transfer behavior, heterojunctions are categorized into three main types (Figure 14): type I, type II, and type III. In type I (straddling gap) heterojunctions, both the conduction band (CB) and the valence band (VB) edges of semiconductor A lie within the band gap of semiconductor B. As a result, both the photogenerated electrons and holes tend to accumulate in semiconductor B, leading to poor spatial separation of the charge carriers and limited redox activity. Type II (staggered gap) heterojunctions exhibit a stepped band alignment, where the CB and VB edges of the two semiconductors are offset such that electrons migrate to the semiconductor with the lower CB and holes to that with the higher VB. This spatial separation improves the charge carrier’s lifetime but significantly reduces the redox potential due to energetic losses during interfacial transfer.

Type III (broken gap) heterojunctions feature no overlap between the band edges, creating an energy barrier that impedes direct electron–hole recombination but also challenges efficient charge transfer and is thus rarely used in photocatalysis.

Z-scheme heterojunctions involve the recombination of photogenerated electrons from the CB of one semiconductor with holes from the VB of another, leaving behind electrons in a more negative CB and holes in a more positive VB, preserving a high redox potential and enabling stronger photocatalytic reactions. However, traditional Z-schemes have often suffered from complex architectures and recombination losses [79]. To overcome these limitations, the S-scheme (step-scheme) heterojunction was proposed as an evolution of the Z-scheme mechanism. It introduces an internal electric field at the interface due to Fermi level equilibration and band bending, which selectively drives the recombination of low-energy charge carriers (i.e., electrons from the less negative CB and holes from the less positive VB) while retaining high-energy electrons and holes in their respective bands. The S-scheme thus combines the advantages of spatial separation and redox potential retention, without the need for mediators, and has shown promising results in many applications, such as water splitting and pollutant degradation applications [80].

Table 2 shows the different strategies developed to form efficient heterojunctions between bismuth-based oxyfluorides and another visible-light active semiconductor (TiO_2_, BiVO_4_, or gC_3_N_4_…), other bismuth-based catalysts (which often induce good material compatibility and simple synthesis through one-pot processes), or metallic nanoparticles (to additionally play with their plasmonic effect).

#### 4.4.1. With Semiconductors and Carbon Materials

The work of H. Razavi-Khosroshahi et al. nicely illustrates how the respective band positions of both semiconductors might drive the material choice in a heterojunction. Indeed, they propose a hydrothermal synthesis method for preparing various BiVO_4_/BiOX heterojunctions (with X = F, Cl, Br, I) and find that BiVO_4_/BiOF shows the best performance for photodegrading methylene blue (Figure 15a) [81]. Compared to other systems, BiVO_4_/BiOF is the only one that forms a type II heterojunction (Figure 15b), allowing electrons to be driven to BiOF and holes to BiVO_4_, reducing their recombination. On the opposite side, in other systems, electrons and holes flow to the same material, BiVO_4_ in BiVO_4_/BiOCl or BiOI in BiVO_4_/BiOCl, increasing their recombination. The BiVO_4_/BiOF association presents as an additional advantage in involving two semiconductors with different band gaps, which increases the range of harvested light. BiVO_4_/BiO_0.67_F_1.66_ cake-like microstructures were successfully prepared through a two-step hydrothermal route by X. Feng et al. [82]. These BiVO_4_/BiO_0.67_F_1.66_ composites demonstrate significantly enhanced photocatalytic activity under visible light for decomposing RhB and phenol in water, compared to that of pure BiVO_4_, due to their increased surface area and the formation of a unique p–n heterojunction.

Nasr et al. synthesized a novel BiOF/TiO_2_ heterostructure and found that the obtained system had a band gap energy of 3.02 eV and could absorb more light and ensure charge separation efficiency (Figure 15c,d) [83]. Qiang et al. designed a series of BiOF/Bi_2_O_3_ heterojunction nanosheets and found that they could effectively separate electron–hole pairs and improve the photocatalytic activity [84]. Moreover, the construction of heterojunction structures with a narrower-band-gap semiconductor such as Ag_2_O strengthens the visible-light response, promotes the separation of the photogenerated charge carriers, and thus enhances the photocatalytic activity [85]. In addition, Hu et al. prepared BiOF/Bi_2_O_3_/RGO hybrid composites and found that the involvement of RGO increased the absorption of light, resulting from a smaller band gap energy [86]. Vadivel et al. fabricated a Ag-BiOF/g-C_3_N_4_ composite and found that the band gap decreased to 2.72 eV, greatly enhancing the absorption of light [87].

**Figure 15 molecules-30-03784-f015:**
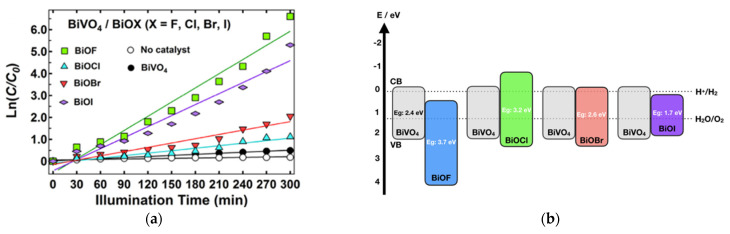
(**a**) The kinetics of the photodegradation of methylene blue over the synthesized samples under visible light per area of particles. (**b**) The band positions of BiVO_4_, BiOF, BiOCl, BiOBr, and BiOI and the respective possible pn junctions [81]. (**c**) The photocatalytic degradation of methylene blue without a photocatalyst (black line) and in the presence of the photocatalysts BiOF (blue line), TiO_2_ (green line), and BiOF/TiO_2_ (1:3) (red line) under visible-light irradiation [83]. (**d**) The proposed mechanism for the photogenerated charge separation and migration process in BiOF/TiO_2_ heterostructures under visible-light irradiation. (**e**) The proposed photocatalytic degradation mechanism for Bi/BiOI_1−x_F_x_ (x = 0.2) under visible-light irradiation [88].

N. Ghayoumia et al. [89] developed BiOF@ZIF-8 (ZIF-8: (Zn(MeIM) 2; MeIM = imidazolate-2-methyl). This heterogeneous compound has been evaluated for the catalytic conversion of the alcoholic functions of benzylic alcohols into carbonyl groups. BiOF@ZIP-8 is very selective towards benzaldehyde (95%). This new catalyst is thus expected to be useful for green oxidation in organic synthesis and low-cost industrial chemical applications [89].

HF wet etching has also been reported as a technique that can improve the photocatalytic properties of bismuth-oxide-based glass thanks to heterojunction creation. Indeed, this type of etching allowed V.P. Singh and R. Vaish to form a BiO_0.51_F_1.98_ layer on SrO-Bi_2_O_3_-B_2_O_3_ glass ceramic and S. Kumar Sharma et al. to form a BiO_0.1_F_2.8_ layer on 2Bi_2_O_3_-Bi_2_O_3_ (BBO) glass. Both were tested for the photodegradation of methyl blue and Rhodamine 6G and showed an enhanced visible-light-driven photocatalytic activity compared to that of unetched glass ceramics [90,91].

Heterojunctions may also be obtained by forming a shell of a Bi_2_NbO_5_F Aurivillius compound around a core of BiF_3_ nanoparticles through a molten salt method. As illustrated by Figure 16, depending on the time and the EG/H_2_O ratio, this core–shell structure can be controlled to accelerate the photodegradation of RhB under visible light [92].

Thus, the formation of heterostructures with other semiconductors can improve both the light absorption range and the carrier separation efficiency, which enhance the photocatalytic activity further.

#### 4.4.2. Other Bismuth Oxyhalides

The use of different bismuth halides for internal coupling not only can avoid the introduction of impurities but can also change the electronic properties, adjust the band gap energy, and enhance the photocatalytic activity. Wang et al. calculated the band gap of a BiOX/BiOY system using hybrid DFT and found out that the band gap appears between BiOX and BiOY [39]. Zhao et al. systematically studied the lattice constants, band gaps, and optical properties of a BiOX_1−x_Y_x_ solid solution, also using DFT calculations [93]. The results revealed that the visible-light response range of the BiOX_1−x_Y_x_ solid solution was wider and the electron–hole recombination rate was much lower [93].

Yan et al. synthesized BiOBr/BiOF hybrid materials and determined that the band gap was significantly narrowed and the photo-absorption and photocatalytic properties were significantly enhanced [94]. Cheng et al. prepared a BiOCl/BiOF composite photocatalyst and found that good band-matching between the oxygen vacancy states in BiOCl and BiOF resulted in effective separation of the electron–hole pairs [95]. A Bi_7_F_11_O_5_/BiOCl heterojunction was prepared using a simple hydrothermal method by Y. Kan et al. In comparison with pure BiOCl and Bi_7_O_5_F_11_, the composite showed an improved photocatalytic activity due to the separation and transfer of the charges by the heterojunction [96]. Guan et al. prepared BiO_0.51_F_1.98_ and calcination polycrystalline BiO_0.51_F_1.98_ with self-doped BiOF through solvothermal synthesis. This material showed enhanced photocatalytic activity, which was basically ascribed to the broadened visible-light absorption and lower recombination rate of the charge carriers [97]. Jingzhen Wang et al. synthesized Bi_7_O_5_F_11_/BiOF and demonstrated its very high performance for the photodegradation of perfluorooctanoic acid. The enhanced performance is attributed to the introduction of a built-in electric field into the S-scheme photocatalyst, providing a strong driving force for the reaction [98]. Jing-Ya Fu et al. prepared BiO_m_F_n_/BiO_x_I_y_/GO with different compositions. Their study noted the excellent photocatalytic activity regarding dye photodegradation of the Bi_50_O_59_F_32_/BiOI/GO composite. Yu-Yun Lin et al. synthesized various quaternary BiOF/BiOI/Bi_26_O_38_F_2_/g-C_3_N_4_ photocatalysts by varying the pH and temperature conditions to optimize their composition. The outstanding photocatalytic performance of BiO_x_F_y_/BiO_p_I_q_/g-C_3_N_4_ is mainly due to the formation of heterojunctions, which enhances the photocatalytic efficiency by reducing the recombination rate of the photogenerated electrons and holes. Additional factors contributing to its performance include the composite’s low-energy band structure, high BET surface area, and layered architecture [49,50].

Aurivillius compounds have also been associated with bismuth oxyhalides. Through hydrothermal synthesis, mixing oxides in a sealed gold tube in NH_4_F 1M medium at a high temperature (≥400 °C) and a high pressure (107 Pa), Bi_2_NbO_5_F and Bi_2_TiO_4_F_2_ phases have been obtained with a small amount of BiOF [99]. The hydrothermal method may also be used to grow Bi_2_TiO_4_F_2_ on a BiOBr (001) sheet and then produce a composite exhibiting a better photocatalytic performance for oxygen production due to the larger charge separation-induced polarization at the heterojunction than that in single materials [100].

#### 4.4.3. Metal Nanoparticles

The integration of metallic nanoparticles, particularly those composed of noble metals such as Au, Ag, Pd, Ru, Rh, and Pt, has emerged as a powerful strategy for enhancing the performance of semiconductor-based photocatalysts. This enhancement is largely attributed to localized surface plasmon resonance (LSPR), a phenomenon wherein collective oscillations in the conduction electrons in metallic nanoparticles are excited by incident light, leading to significant amplification of the local electromagnetic field near the nanoparticle surface. This effect induces strong absorption and scattering of light in the visible range. This plasmonic effect also promotes the generation of energetic “hot electrons”. These hot electrons can be injected into the conduction band of the adjacent semiconductor, effectively boosting the charge carrier population and improving the photocatalytic efficiency [101,102]. Furthermore, the presence of metallic nanoparticles can modulate the local Fermi level equilibrium and facilitate the formation of heterojunctions, which enhance the charge separation and reduce electron–hole recombination [103,104]. In some systems, metallic nanoparticles serve a dual function by acting as cocatalysts, providing active sites for redox reactions [105]. Additionally, the incorporation of metallic nanoparticles into Z-scheme photocatalytic architectures allows for spatially separated oxidation and reduction reactions while maintaining strong redox potentials. Beyond purely electronic effects, the plasmonic heating induced by LSPR can also contribute to photothermal catalysis, where localized temperature rises at the nanoparticle–semiconductor interface accelerate the reaction kinetics and facilitate thermally activated surface processes [106]. Overall, the multifunctional roles of metallic nanoparticles in modifying the electronic structure, enhancing light–matter interactions, and facilitating the interfacial charge dynamics represent a promising approach to the design of new photocatalysts. S. Vadivel et al. synthesized both Ag-BiOF/g-C_3_N_4_ and BiOF/g-C_3_N_4_ through a solvothermal method and showed that the Ag-modified BiOF specimen had a significantly enhanced photocatalytic activity for the degradation of organic pollutants. The increased photocurrent measured confirms the improved separation of the photogenerated charges. Finally, the decreased band gap, to 2.72 eV, greatly enhanced the absorption of light [87].

The plasmon resonance of non-noble metals (Al, Cu, Mg, In, Ni, Ga, Co, Fe, Bi) has also been investigated and could be of interest due to their much lower manufacturing cost potential [107]. Metal Bi exhibits an SPR that is similar to that of noble metals, and recently, Bi nanoparticles have been used to upgrade the photocatalytic properties of many semiconductors, such as BiVO_4_, TiO_2_, and Bi_2_Sn_2_O_7_ [108,109,110]. L. Ai et al. prepared a Bi/BiOF/Bi_2_O_2_CO_3_ Z-scheme heterojunction. The enhanced photocatalytic activity for ciprofloxacin degradation can be attributed to use of metal Bi, which extends the light absorption range and whose SPR improves the rate of transfer of the photogenerated electron–hole pairs to the semiconductor’s surface. J. Wang et al. synthesized a Bi/BiOI_1−x_F_x_ solid solution and demonstrated that compared with pure BiOI and BiOF, the band gap of the metal–oxyhalide compound was significantly reduced, thus promoting visible-light absorbance. Figure 15e shows the photocatalytic mechanism of perfluorooctanoic acid degradation [88]. S. Ibrahim et al. deposited BiO_0.5_F_2_ through reactive magnetron sputtering in different Ar/O_2_/CF_4_ gas mixtures. By modulating Rf (O_2_/(O_2_ + CF_4_)), it was possible to synthesize one-pot heterojunctions of BiO_0.5_F_2_/Bi with a controlled content of both phases. These thin films were tested for the photoconversion of CO_2_ into CO and compared with a Bi_2_O_3_/Bi heterojunction thin film, Bi_2_O_3_ powder, and BiOF powder. The BiO_0.5_F_2_/Bi heterojunction shows both the best photocatalyst yield and selectivity for CO [111,112].

**Table 2 molecules-30-03784-t002:** Synthesis methods, precursors, and photocatalytic applications of BiO_x_F_3−2x_-based materials.

Materials	Synthesis Method	Precursors	Application	References
Ag-BiOF/g-C_3_N_4_	Solvothermal	g-C_3_N_4_, Bi(NO_3_)_3_·5H_2_O, AgNO_3_, C_6_H_12_N_4_, NaF, C_2_H_6_O_2_	Photodegradation of MB under visible light	[87]
Bi/BiOF/Bi_2_O_2_CO_3_	Hydrothermal	Bi(NO_3_)_3_·5H_2_O, NaF, C_2_H_6_O_2_/H_2_O (1:1)	Photodegradation of ciprofloxacin under UV-Vis light	[112]
Bi/BiOI_1−x_F_x_	Hydrothermal	Bi(NO_3_)_3_·5H_2_O, NaF, KI, C_2_H_6_O_2_	Photodecomposition of perfluorooctanoic acid	[88]
BiO_0.5_F_2_/Bi	Magnetron sputtering	Bi target, Ar/O_2_/CF_4_ atmosphere	Photodegradation of MO	[111]
Ag_2_O/BiOF	Solvothermal	Bi(NO_3_)_3_·5H_2_O, NH_4_F	Photodegradation of RhB under UV-Vis light	[85]
BiOF/Bi_2_O_3_/RGO	Precipitation/calcination/reduction	Bi(NO_3_)_3_·5H_2_O, NaF, GO	Photodegradation of RhB under natural sunlight	[86]
BiVO_4_/BiO_0.67_F_1.66_	Two-step hydrothermal synthesis	Bi(NO_3_)_3_·5H_2_O, C_2_H_6_O_2_, NaVO_3_, NH_4_F	Photodegradation of RhB and phenol molecules	[82]
BiVO_4_/BiOF	Hydrothermal	Bi(NO_3_)_3_·5H_2_O, C_2_H_6_O_2_, Na_3_VO_4_, NH_4_F	Photodegradation of MB under visible light	[81]
BiOF/TiO_2_	Solid-state and sintering	Bi_2_O_3_, HF, TiO_2_	Photodegradation of RhB and MB under visible light	[83]
BiOF@ZIF-8	Hydrothermal	Bi(NO_3_)_3_·5H_2_O, terephthalic acid, NH_4_F, HF, Zn(NO_3_)_2_·6H_2_O, H-MeIM, DMF, CHCl_3_, DCM	Oxidation of benzylalcohols	[89]
BiOF/Bi_2_O_3_	Hydrothermal	Bi(NO_3_)_3_·5H_2_O, HF/NH_3_·H_2_O	Photodegradation of RhB under UV–visible light	[84]
BiO_0.51_F_1.98_ coated SrO-Bi_2_O_3_-B_2_O_3_ glass ceramic	Fluorination through controlled etching	SBBO, HF	Photodegradation of MB under visible light	[91]
BiO_0.1_F_2.8_ coated 2Bi_2_O_3_-Bi_2_O_3_ (BBO) glass	Wet etching	BBO, HF	Photodegradation of Rh6G dye	[90]
BiOBr/BiOF	Solvothermal	CTAB, NaF, HF, Bi(NO_3_)_3_·5H_2_O, C_2_H_6_O_2_	Photodegradation of MB under visible light	[94]
BiOBr/BiOF	Hydrothermal	CTAB, NH_4_F, Bi(NO_3_)_3_·5H_2_O	Photodegradation of RhB and nitrobenzene under visible light	[113]
BiOCl/BiOF	Solvothermal	BiCl_3_, NH_4_F, Bi(NO_3_)_3_·5H_2_O, C_4_H_10_O_2_	Photocatalytic degradation of methyl orange (MO) under UV–Vis.-light irradiation	[95]
BiO_0.51_F_1.98_/BiOF	Solvothermal	NaF, Bi(NO_3_)_3_·5H_2_O, C_2_H_6_O_2_	MO degradation under light irradiation	[97]
Bi_7_O_5_F_11_/BiOF	Solvothermal	NH_4_F, Bi(NO_3_)_3_·5H_2_O, C_2_H_6_O_2_, H_2_O	Photodegradation of PFOA under UV-light irradiation	[98]
Bi_7_F_11_O_5_/BiOCl	Hydrothermal	BiOCl, NH_4_F, C_2_H_6_O/H_2_O	Photodegradation of MO under UV light	[96]
BiO_m_F_n_/BiO_x_I_y_/GO	Hydrothermal	C Graphite, NaNO_3_, H_2_SO_4_., KMnO_4_, Bi(NO_3_)_3_·5H_2_O, NaOH, KI, KF	Photodegradation of CV and HBA under visible light	[51]
BiOF/BiOI/Bi_26_O_38_F_2_/g-C_3_N_4_	Hydrothermal	Melamine, Bi(NO_3_)_3_·5H_2_O, NaOH, HNO_3_, KF, KI	Photodegradation of CV under visible light and photoconversion of CO_2_ into CH_4_ under visible light	[49]
Bi_2_TiO_4_F_2_/BiOBr	Solvothermal	Bi(NO_3_)_3_·5H_2_O, (NH_4_)_2_TiF_6_, C_2_H_6_O_2_	Photocatalytic O_2_ evolution	[100]
BiF_3_/Bi_2_NbO_5_F	Solvothermal	Bi(NO_3_)_3_·5H_2_O, Nb_2_O_5_, C_2_H_6_O_2_	Photodegradation of RhB under visible-light irradiation	[92]

### 4.5. The Use of Sensitizers

The use of dye sensitizers can increase the absorption of visible light and enrich the photogeneration of electric charge carriers. Yadav et al. synthesized quercetin-sensitized BiOF nanostructures using a non-toxic and economical route. The electron acceptor properties of quercetin promoted the charge injection into BiOF, as shown in Figure 17, resulting in higher light absorption, a narrower BiOF band gap, and a larger specific surface area [114].

## 5. Conclusions

This paper focuses on bismuth-based oxyfluorides as emergent photocatalysts, highlighting their potential to address energy shortages and environmental pollution through photocatalysis. While titanium dioxide (TiO_2_) has been widely used, it has disadvantages such as low sunlight utilization and quantum efficiency. Recently, bismuth-based materials have garnered significant attention. Bismuth oxyfluoride (BiOF) stands out among these due to its unique layered structure, the high electronegativity of fluorine (which enhances the photocatalytic efficiency by trapping electrons and influencing the electron distribution), and its direct band gap, which facilitates efficient light absorption.

Beyond pure BiOF, BiO_x_F_3−2x_ compounds with finely varied stoichiometries have started to be investigated, as their O/F ratios influence oxygen defects and photocatalytic activity. Furthermore, Aurivillius bismuth oxyfluorides, which present layered perovskite-like structures, are gaining attention due to their unique electronic functionalities and ferroelectric properties, making them promising for photocatalytic applications. This large variety of compositions allows them to adapt to a targeted redox potential for a specific photocatalytic application.

Moreover, several strategies can be explored for enhancing the photocatalytic activity of these pure materials. First, the facet selection and oxygen vacancies play a critical role, as oxygen vacancies can improve electron–hole separation and enhance light absorption. Metal/non-metal doping is also a common approach to adjusting the band gap and accelerating the charge separation in a photocatalyst. Rare earth ions (e.g., Sm^3+^, Y^3+^, Er^3+^/Yb^3+^, Eu^3+^) and noble metals (e.g., Ag, Pd) were demonstrated to be effective dopants for enhancing the photocatalytic activity under visible or NIR light. Moreover, heterostructure formation with other semiconductors (e.g., TiO_2_, BiVO_4_, Bi_2_O_3_, g-C_3_N_4_), other bismuth-based catalysts (e.g., BiOBr, BiOCl), and metal nanoparticles (e.g., Ag, Bi) is a highly effective strategy for reducing electron–hole recombination and extending light absorption. Finally, a few papers have indicated the use of sensitizers, such as quercetin, which can also significantly increase visible-light absorption and charge carrier photogeneration.

Environmental remediation and solar fuel production are then two important applications of photocatalysis that can be addressed by BiOF and its derivatives. In addition, theoretical calculations should be considered in future studies, as they could provide fundamental insights into the photocatalytic mechanisms of bismuth oxyfluorides and guide the rational design of more efficient materials. However, from a practical point of view, it remains difficult to recycle a powder catalyst from liquid and requires the addition of a filtration step. The use of thin films could overcome this problem and lead to reusable and ecofriendly systems. However, up to now, no studies concerning BiOF-based thin films have been published, and very few on BiO_x_F_3−2x_ materials have been published.

## Figures and Tables

**Figure 1 molecules-30-03784-f001:**
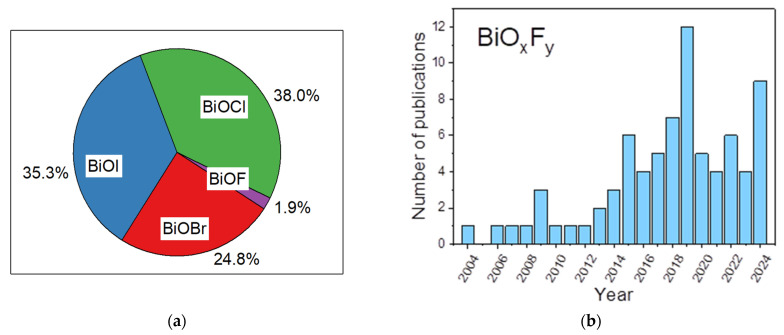
As per Web of Science data, (**a**) pie chart of articles dealing with tetragonal matlockite (PbFCl-type) oxyhalides BiOBr, BiOCl, BiOI, and BiOF by percentage; (**b**) number of publications on “Bismuth oxyfluoride”, “BiOF”, “Bi_7_O_5_F_11_”, “BiO_0.5_F_2_”, “BiO_0.51_F_1.98_”,“BiO_0.67_F_1.66_”, “BiO_1.18_F_0.64_”, or “Bi_50_O_59_F_32_” from 2004 to 2024 (Web of Science, 31 December 2024).

**Figure 2 molecules-30-03784-f002:**
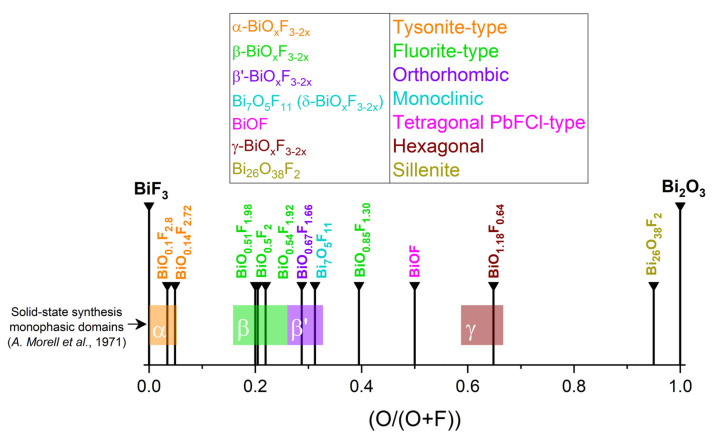
Known BiO_x_F_3−2x_ compositions and related phases [35].

**Figure 3 molecules-30-03784-f003:**
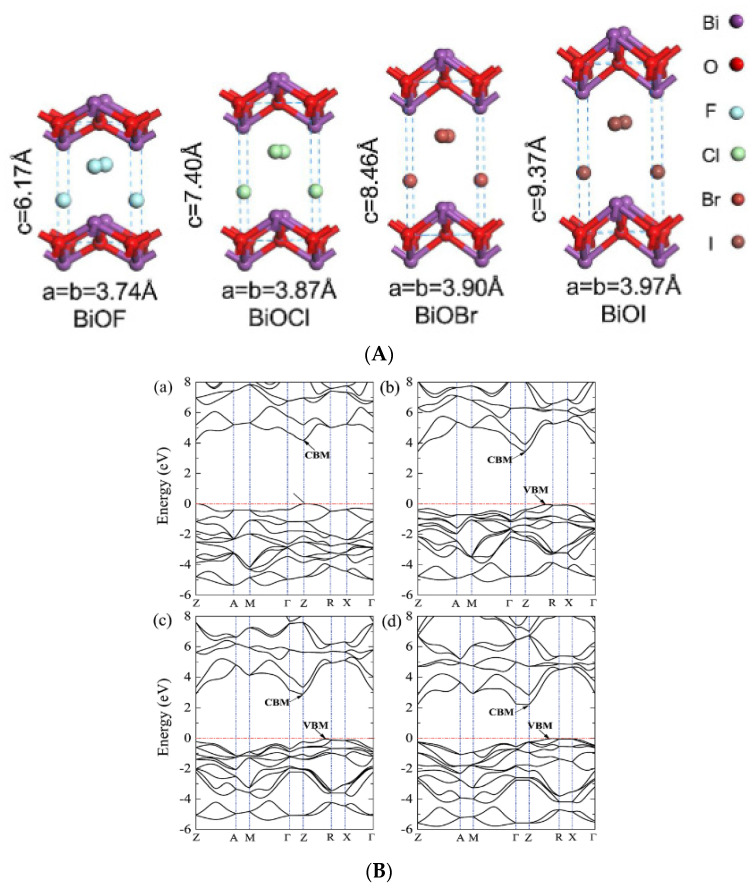
(**A**) A schematic diagram of BiOX’s crystal structure [37]; (**B**) the band structures of (**a**) BiOF, (**b**) BiOCl, (**c**) BiOBr, and (**d**) BiOI. The horizontal dashed lines represent the Fermi levels [39].

**Figure 4 molecules-30-03784-f004:**
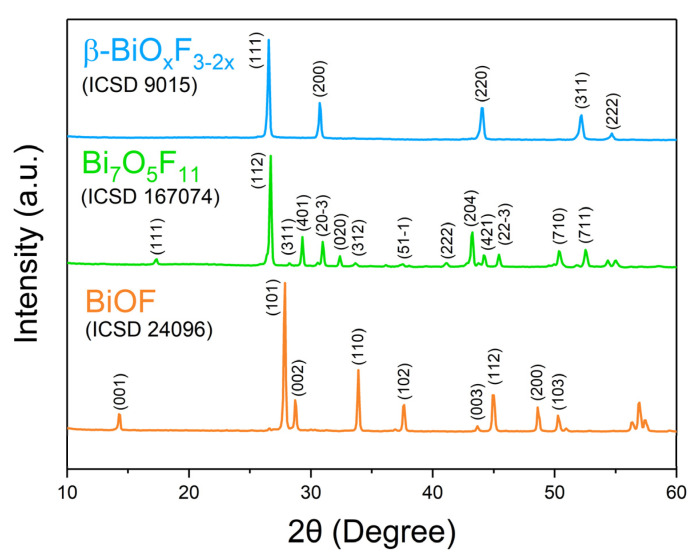
XRD patterns of BiOF, Bi_7_O_5_F_11_, and β-BiO_x_F_3−2x_ [40].

**Figure 5 molecules-30-03784-f005:**
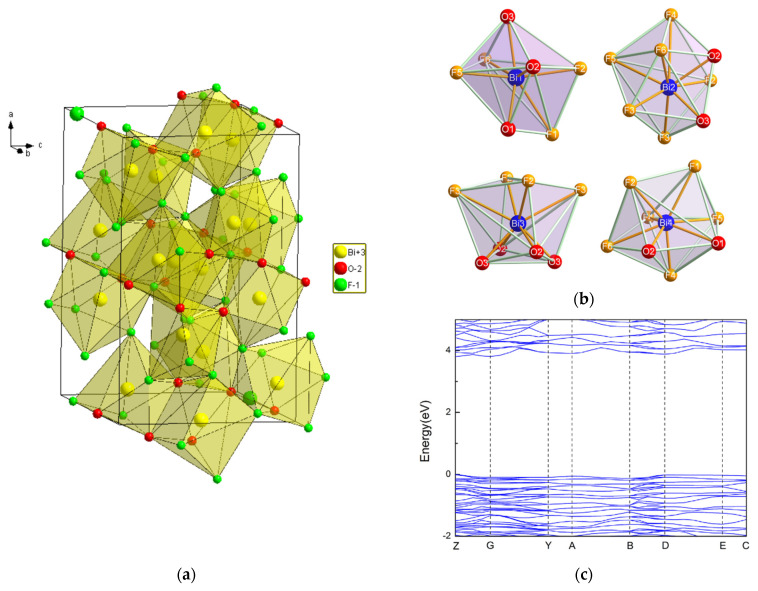
(**a**) A perspective view of the hexagonal Bi_7_O_5_F_11_ bismuth oxyfluoride structure; (**b**) schematic representations of the various polyhedra around the Bi atoms in Bi_7_O_5_O_11_ [43]; (**c**) the band structure of Bi_7_O_5_O_11_ [43].

**Figure 6 molecules-30-03784-f006:**
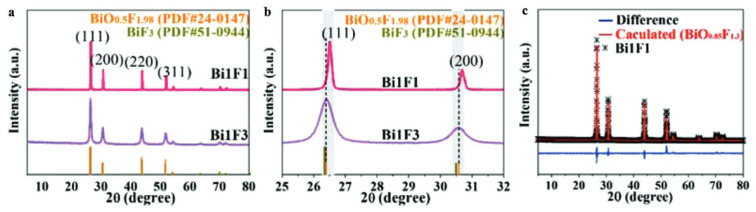
(**a**,**b**) XRD patterns of Bi1F3 (BiO_0.54_F_1.92_) and Bi1F1 (BiO_0.85_F_1.3_). (**c**) Rietveld refinement of XRD pattern of Bi1F1 [44].

**Figure 7 molecules-30-03784-f007:**
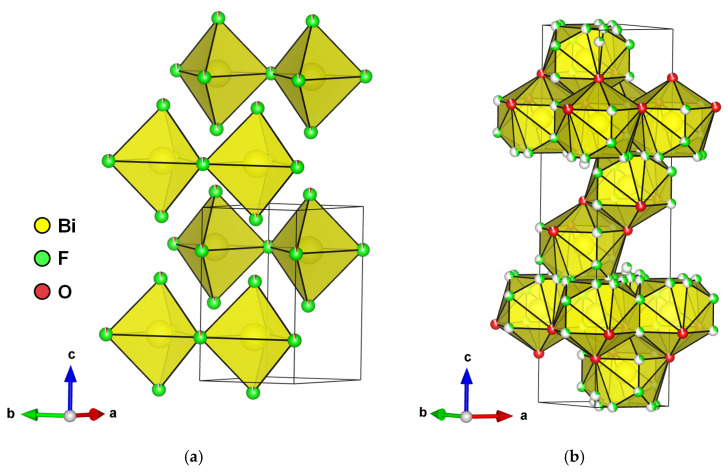
(**a**) Perspective view of hexagonal BiO_1.18_F_0.64_ bismuth oxyfluoride structure and (**b**) rhombohedral BiO_0.9_F_2.35_ bismuth oxyfluoride.

**Figure 8 molecules-30-03784-f008:**
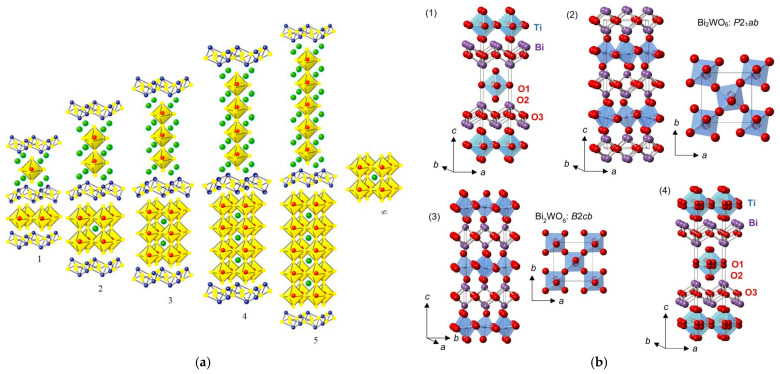
(**a**) An Aurivillius structure as a function of the number of pseudo-perovskite layers [52]. (**b**) An illustration of (**1**) the ideal ordered *I*4/*mmm* structure for an *n* = 1 Aurivillius phase, (**2**) a low-temperature P2_1_*ab* phase for Bi_2_WO_6_, (**3**) an intermediate *B*2*cb* phase for Bi_2_WO_6_, and (**4**) a disordered *I*4/*mmm* structure of Bi_2_TiO_4_F_2_ (with displacive disorder of the equatorial and apical anion positions to the 16_n_ and 16_m_ sites, respectively) from 100 K NPD Rietveld refinement; TiX_6_ or WO_6_ polyhedra, Bi, and O are shown in blue, purple, and red, respectively [53,54].

**Figure 9 molecules-30-03784-f009:**
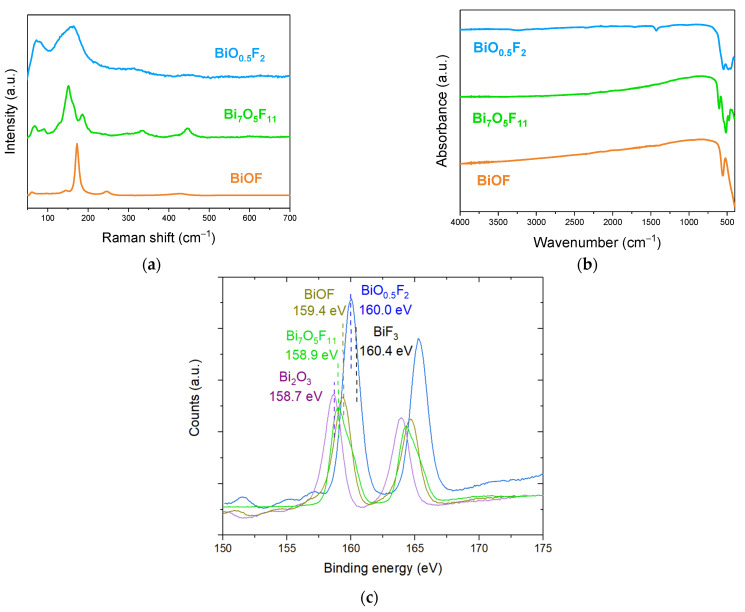
(**a**) Raman and (**b**) FTIR spectra of BiOF, Bi_7_O_5_F_11_, and β-BiO_0.5_F_2_; (**c**) XPS spectra of Bi 4f from Bi_2_O_3_, BiOF, Bi_7_O_5_F_11_, and β-BiO_0.5_F_2_ and BiF_3_ compounds [40].

**Figure 10 molecules-30-03784-f010:**
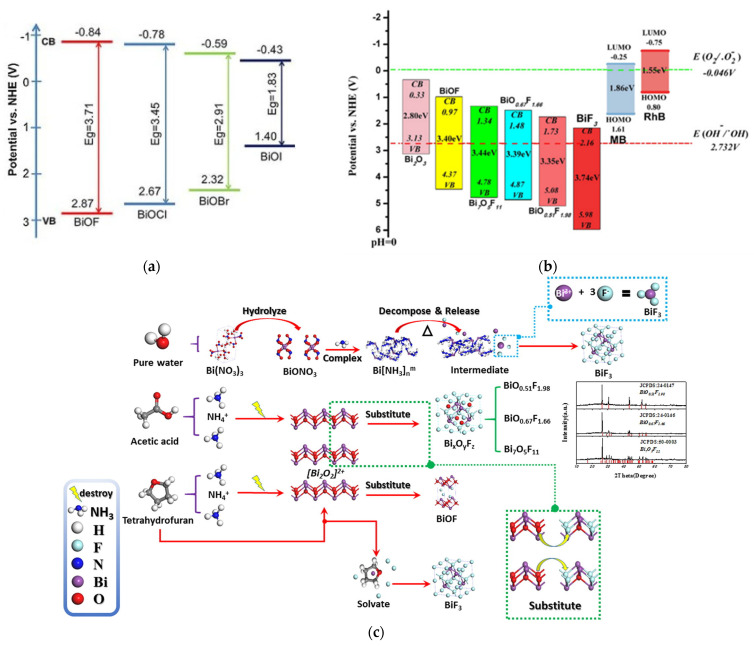
The redox potentials of photocatalytic reactions with respect to the estimated flat band edge positions of (**a**) various BiOX [37] and (**b**) Bi_x_O_y_F_z_ [45] specimens. (**c**) The plausible conversion mechanism of Bi_x_O_y_F_z_: in acetic acid/water conditions, acetic acid would hydrolyze to produce H^+^. H^+^ and NH_4_^+^ as a Lewis acid may partially destroy the [Bi_2_O_2_]^2+^ layer, where the O^2−^ in the [Bi_2_O_2_]^2+^ layer will be substituted with F^−^, leading to the formation of BixOyFz with different O/F ratios [45].

**Figure 11 molecules-30-03784-f011:**
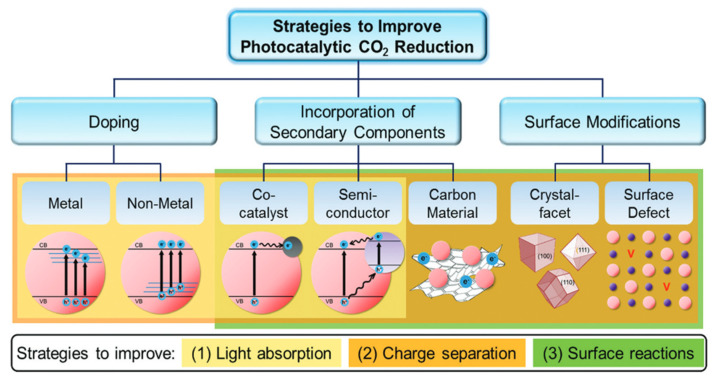
Strategies for an advanced photocatalytic CO_2_ reduction performance by improving the light absorption ability, charge separation efficiency, and surface reactions [72].

**Figure 12 molecules-30-03784-f012:**
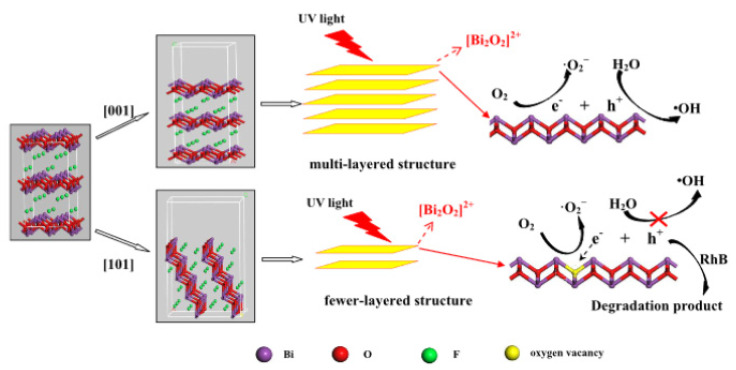
A schematic of the reaction mechanisms of multi- and few-layered structures [27].

**Figure 13 molecules-30-03784-f013:**
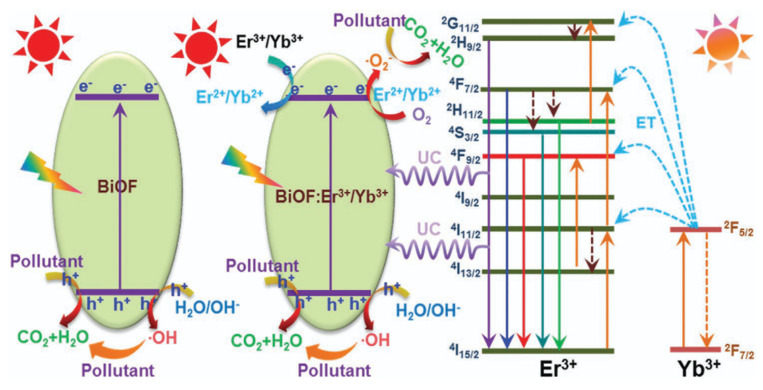
A schematic diagram of the possible photocatalytic mechanisms of the designed nanoparticles under NIR and visible-light irradiation [76].

**Figure 14 molecules-30-03784-f014:**
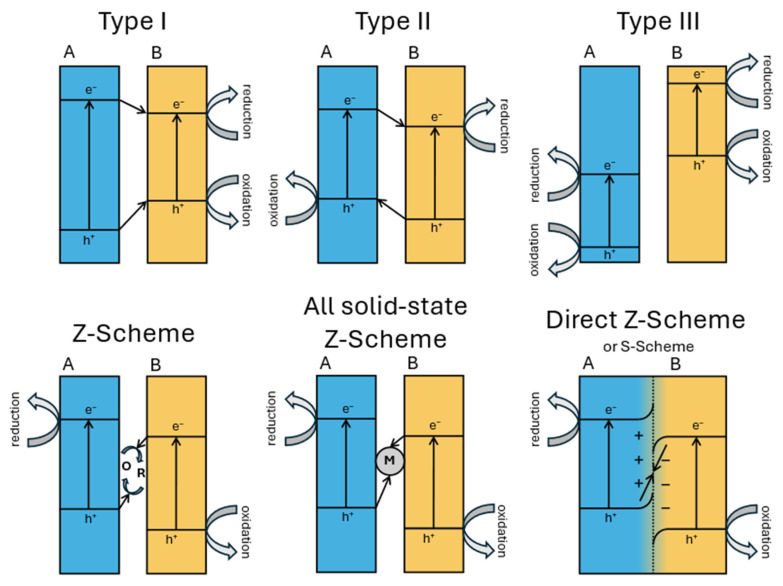
A schematic representation of band diagrams of different types of heterojunctions.

**Figure 16 molecules-30-03784-f016:**
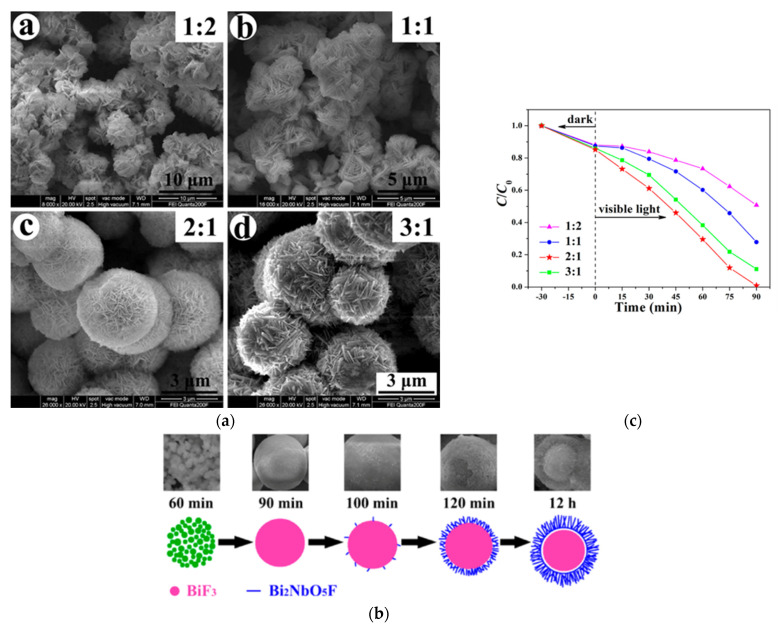
(**a**) Morphological evolution as a function of the EG/H_2_O ratio for BiF_3_/Bi_2_NbO_5_F heterojunction formation; (**b**) the proposed mechanism of this synthesis; (**c**) the photocatalytic activity of these structures as a function of the EG/H_2_O ratio [92].

**Figure 17 molecules-30-03784-f017:**
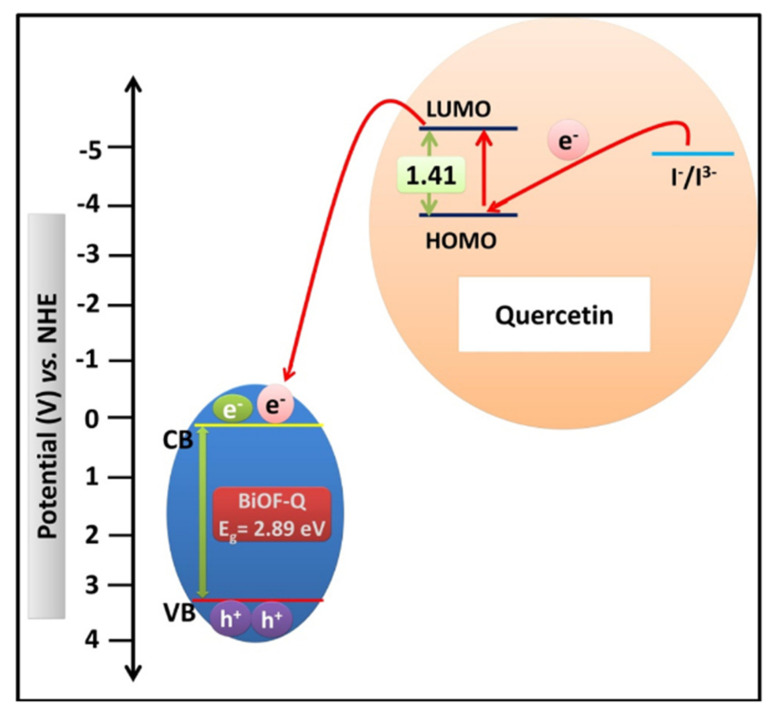
A schematic illustration of an energy-level diagram displaying the injection of electron from the quercetin molecule into the CB of BiOF-Q and the subsequent regeneration of the quercetin molecule [114].

## Data Availability

The data is contained within the article.

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
