# Peer review of "Bismuth-Based Oxyfluorides as Emergent Photocatalysts: A Review"

_molecules, 2025, doi:10.3390/molecules30183784_

Round 1
Reviewer 1 Report
Comments and Suggestions for Authors
35-39 "When the photocatalyst is irradiated with photon..."
It should be clarified that this is only true for semiconductor photocatalysts
Figure 1 b
The quality of the illustration is low. And in general there is no need for a three-dimensional representation, it is more convenient to represent it in the form of sectors of a circle.
Figure 2
The illustration is not clear, what the colored horizontal stripes mean needs to be labeled.
Figure 12
It would be more convenient for the reader to provide the potentials for both parts a and b on a single scale. And also add potential for carbon dioxide conversion
Figure 13 and 15
Needs more detailed explanation and comments
Reviewer 2 Report
Comments and Suggestions for Authors
The authors stated: “As shown in Figure 1a, the publications on bismuth oxyfluoride photocatalysts have increased yearly since 2013, revealing the scientific community’s interest in BiOxF3-2x as a photocatalyst.” However, from Figure 1 it is evident that the publication trend on BiOF is not consistent over time, and only a limited number of articles have been published. It is unclear why the authors chose to highlight this material.
The authors report that BiOF has a bandgap of approximately 3.7 eV, placing it in the ultraviolet light-active region. The manuscript should provide a clear justification for the relevance and potential interest of this material despite its wide bandgap.
From Figure 12, it appears that BiOX materials strongly support oxidation processes, and BiOF, in particular, exhibits higher oxidation potentials. This property could be useful in applications requiring high oxidation potentials and may serve as one of the key points the authors could emphasize in the discussion.
In addition, the circle between the two semiconductors in the Z-scheme diagram is misleading and requires careful correction. The current figure suggests an all-solid-state Z-scheme, which differs from a direct Z-scheme (or S-scheme), where no metal is present between the two semiconductors. Therefore, the figure should be revised carefully to provide all the necessary information or, alternatively, simplified to directly represent the S-scheme.
Finally, Figure 17 should include sub-labeling, such as (a), (b), etc.
Reviewer 3 Report
Comments and Suggestions for Authors
This paper focuses on Bismuth-based oxyfluorides as emergent photocatalysts, high- 646 lighting their potential in addressing energy shortage and environmental pollution 647 through photocatalysis. While titanium dioxide (TiO2) has been widely used, it has disadvantages such as low sunlight utilization and quantum efficiency. Recently, bismuth based materials have garnered significant attention. Bismuth oxyfluoride (BiOF) stands out among these due to its unique layered structure, the high electronegativity of fluorine (which enhances photocatalytic efficiency by trapping electrons and influencing electron distribution), and its direct band gap that facilitates efficient light absorption.
- The magnification of SEM image is a bit low, making it difficult to see the microstructure of the material clearly.
- Thequality of the picture needs to be improved, Some figures should be merged together.
- S-schemeheterojunction photocatalysts should be elaborated.
- Theoretical calculations are important for the mechanism explanation of photocatalysis, should examples be given?
The innovativeness of this review should be given in detail in the introduction.
Round 2
Reviewer 2 Report
Comments and Suggestions for Authors
accept